Description and rediagnosis of the crested hadrosaurid (Ornithopoda) dinosaur Parasaurolophus cyrtocristatus on the basis of new cranial remains

Gates Terry A. 1 2 3 tagates@ncsu.edu
http://orcid.org/0000-0001-9608-6635 Evans David C. 4
http://orcid.org/0000-0001-8096-3605 Sertich Joseph J.W. 5
1 Department of Biological Sciences, North Carolina State University , Raleigh, NC , USA
2 Paleontology Unit, North Carolina Museum of Natural Sciences , Raleigh, NC , USA
3 Department of Geology, Field Museum of Natural History , Chicago, IL , USA
4 Department of Natural History, Royal Ontario Museum , Toronto, ON , Canada
5 Department of Earth Sciences, Denver Museum of Nature & Science , Denver, CO , USA
Knoll Fabien
Electronic publication date: 2021 Jan 25
Publication date: 2021
Volume: 9
Electronic Location ID: e10669
Received 2020 Oct 13; Accepted 2020 Dec 8
Copyright: © 2021 Gates et al.
Copyright year: 2021
Copyright holder: Gates et al.
License: This is an open access article distributed under the terms of the Creative Commons Attribution License, which permits unrestricted use, distribution, reproduction and adaptation in any medium and for any purpose provided that it is properly attributed. For attribution, the original author(s), title, publication source (PeerJ) and either DOI or URL of the article must be cited.
License URL: https://creativecommons.org/licenses/by/4.0/

Keywords: Ontogeny, Cretaceous, Dinosaur, Allometry, Phylogeny, Taxonomy, Campanian, Crest, Sexual selection

Funding: Denver Museum of Nature and Science National Science and Engineering Research Council of Canada Discovery Grant RGPIN-2018-06788 This work was supported by the Denver Museum of Nature and Science and by a National Science and Engineering Research Council of Canada Discovery Grant (RGPIN-2018-06788). The funders had no role in study design, data collection and analysis, decision to publish, or preparation of the manuscript.

==============================
For nearly 60 years, skulls of Parasaurolophus species have been differentiated primarily on the basis of crest shape rather than on unique morphologic characters of other cranial elements. Complicating matters is the fact that crests dramatically change shape throughout ontogeny. Without a complete growth series, it has become difficult to assess the taxonomic distinctness of each species through the lens of allometric growth. Parasaurolophus cyrtocristatus has proven to be especially troublesome to assess because of the poorly preserved nature of the type and only skull. A new, partial skull from the Fossil Forest Member of the Fruitland Formation—the same geologic unit as the type specimen—is the first opportunity to re-diagnose this species as well as redefine the genus with many new traits. An undescribed, short-crested subadult skull from the Kaiparowits Formation of Utah previously assigned to cf. P. cyrtocristatus allows detailed comparisons to be made between the unnamed Utah taxon and the material of this species from the type locality. We find that several characteristics of the squamosal, supraoccipital, and premaxilla shared between the referred skull and the type skull are unique to P. cyrtocristatus (senso stricto) within the genus, irrespective of the overall crest shape. A phylogenetic analysis that includes six new characters posits that P. cyrtocristatus and P. tubicen are sister taxa, and that the latter does not share a closest common ancestor with the long-crested P. walkeri as previously hypothesized. This result helps to explain why both taxa are found in northeastern New Mexico, USA and in sequential geologic units (Fruitland Formation and Kirtland Formation, respectively). Additionally, the exquisitely preserved new skull provides the first opportunity to unequivocally identify the osteological make-up of the Parasaurolophus cranial crest. Unlike in previous reconstructions, the crest composition in Parasaurolophus follows what is seen in other lambeosaurines such as Corythosaurus, where the dorsal process of the premaxilla dominates the crest, with the nasal forming 80% of the ventral paired tubes, and the lateral premaxillary process acting a lateral cover between the dorsal and ventral tubes. The skull of P. cyrtocristatus is still incompletely known, so more complete material will likely reveal new features that further differentiate this species and aid in determining the pace of ornamental crest evolution.

Introduction

Parasaurolophus is a genus of duck-billed dinosaur that is most strikingly characterized by a tubular crest that extends over the top of the skull and beyond the occiput. Internally, the crest is hollow, housing the nasal cavity that has a shape previously described as simple U-shaped tubes with blind tubular diverticulae (Weishampel, 1997; Sullivan & Williamson, 1999). Three species of Parasaurolophus have been described and are currently recognized as valid, all found in the Campanian (Late Cretaceous) of western North America (Fig. 1). Parasaurolophus walkeri (Parks, 1922) is known from a single complete skull and almost complete skeleton from the Dinosaur Park Formation (~77–76 Ma; Eberth, 2005 sensu Fowler, 2017) of Alberta, Canada, and several isolated braincases (Evans, Reisz & Dupuis, 2007; Evans, Bavington & Campione, 2009). Two-thousand kilometers south, in the Kirtland Formation (~75–73.3 Ma; Fassett & Heizler, 2017) of New Mexico, the crest and partial skull of P. tubicen was discovered (Wiman, 1931). A more complete skull of this species is now known (Sullivan & Williamson, 1999). Unlike P. walkeri and P. tubicen, which have long slightly curving crests, Parasaurolophus cyrtocristatus is unique in having a shorter crest that curves sharply ventrally over the posterior end of the skull roof. Ostrom (1961) named the species from a single (FMNH P-27393) partial, poorly preserved skull with an associated, reasonably complete and well-preserved skeleton discovered in the Fruitland Formation (~76.1–75 Ma; Fassett & Heizler, 2017) of New Mexico by Charles Sternberg in 1923 (Ostrom, 1961).

Figure 1 Holotype skulls of the three currently diagnosed species within the Parasaurolophus clade.

(A) Parasaurolophus cyrtocristatus (FMNH P-27393); (B) Parasaurolophus walkeri (ROM 768); and (C) Parasaurolophus tubicen (NMMNH P-25100). Ages associated with each skull represent the approximate timing of each taxon.

Though poorly preserved, the type and only previously known specimen of P. cyrtocristatus possesses the distinct crest morphology that easily differentiated the species from either P. walkeri or P. tubicen, yet this trait also was the sole source of evidence for suggesting sexual dimorphism within the genus (Hopson, 1975; Williamson, 2000). A consequence of the sexual dimorphism hypothesis is that if FMNH P-27393 was a sexual dimorph of either of the two long crested species, that would mean that the name P. cyrtocristatus would ergo be a junior synonym of a different species. Authors have rebuked this hypothesis based on stratigraphic and geographic separation of species (Sullivan & Williamson, 1999; Lund & Gates, 2006; Gates et al., 2007) leaving P. cyrtocristatus as a valid species.

From a taxonomic perspective, the problem with Parasaurolophus cyrtocristatus is that the specific diagnosis is based mostly on crest shape, which is known to change in all lambeosaurines through ontogeny (Dodson, 1975; Evans, Forster & Reisz, 2005; Evans, 2010; Brink et al., 2014), but clearly differs from the other Parasaurolophus species based on the fossil material currently available. Other proposed diagnostic characteristics of this species are generally proportional differences of the postcranial skeleton (Brett-Surman, 1989). Repercussions of using the ambiguous, ontogenetically variable characteristics of crest size and shape is illustrated by the plethora of Parasaurolophus material that has arisen from the Kaiparowits Formation of southern Utah (~77–74 Ma, Roberts et al., 2013 sensu Fowler, 2017). Beginning in 1979, partial skulls of Parasaurolophus were discovered, mostly attributed to P. cyrtocristatus because they all possessed a short curved crest (Weishampel & Jenson, 1979; Sullivan & Williamson, 1999; Lund & Gates, 2006; Gates et al., 2007), whereas other studies have stopped short of attributing a species designation because subsequent discoveries of both juvenile specimens and more mature individuals have shown that the crest changes dramatically through growth (Farke et al., 2013; Gates et al., 2013). Overall, the Parasaurolophus material from the Kaiparowits Formation remains unnamed until a proper diagnosis of P. cyrtocristatus can be resolved.

A new, exquisitely preserved, partial skull of Parasaurolophus cyrtocristatus (DMNH EVP.132300) discovered in the Fossil Forest Member of the Fruitland Formation (Fig. 2) provides the first opportunity to define new species-specific diagnostic skull traits that are more robust to ontogenetic change than crest shape and present across the hypodigm. Results from this work are imperative to delineating the standing diversity of dinosaur species within the Parasaurolophus clade as well as unraveling the phylogenetic relationships among these iconic dinosaurs.

Figure 2 Map of the southern San Juan Basin.

(A) Exposures of the Upper Cretaceous Fruitland and Kirtland formations across northwestern New Mexico and (B) approximate locations of the holotype (FMNH P-27393) and referred (DMNH EPV.132300) specimens of Parasaurolophus cyrtocristatus in the Fossil Forest Member of the Fruitland Formation. Map credit for A: “A new species of trionychid turtle from the Upper Cretaceous (Campanian) Fruitland Formation of New Mexico, USA” COPYRIGHT: © 2018, The Paleontological Society ‘Reprinted with permission’.

Site geology

Hadrosaurid fossils are among the most abundant large vertebrate remains from the upper Fruitland and lower Kirtland formations. As noted by Joyce, Lyson & Sertich (2018), fossils from this interval have traditionally been grouped into the “Hunter Wash Local Fauna,” the fossiliferous horizons in the upper 12.2 meters of the Fruitland Formation and the lower 16.8 m of the Kirtland Formation in the Hunter Wash area (Clemens, 1973; Sullivan & Lucas, 2003, 2006).

The contact between the Fruitland and Kirtland formations was characterized originally as gradational, with sandier sediment in the Fruitland Formation compared to the overlying Kirtland Formation (Bauer, 1916). Later, Fassett & Hinds (1971) defined the contact as the top of the highest coal or carbonaceous shale, providing the foundation for more recent research (Fassett, 2000, 2010). The trouble is that the gradational shift between formations makes the definition geologically problematic.

Other attempts to define the contact (Hunt & Lucas, 1992; Lucas, Hunt & Sullivan, 2006) used the base of the “Bisti Member” or “Bisti Bed” sandstone, as the top of the Fruitland Formation or the base of the overlying Farmington Sandstone Member (“Farmington Member”). Using either definition, the new P. cyrtocristatus locality (DMNH loc. 7047) falls within the Fruitland Formation (Fig. 2). Ash beds bracket the locality, one in the lower Fruitland Formation (“Dog Eye Pond” (DEP) ash of Fassett & Steiner (1997)) whereas the upper ash is found within the lower Kirtland Formation (Ash 2 of Fassett & Steiner (1997)). Recent re-dating of these ashes produced dates of 76.14 ± 0.12 Ma and 75.02 ± 0.04 Ma, respectively (Fassett & Heizler, 2017). These dates contrast to those recalculated by Roberts et al. (2013) and Fowler (2017) using updated standards (DEP = 75.16 ± 0.41 Ma and Ash 2 = 74.17 ± 0.13 Ma; DEP = 76.029 ± 0.41 Ma and Ash 2 = 75.023 ± 0.13 Ma, respectively).

The new Parasaurolophus skull was recovered from the base of a white, cross-bedded sandstone within the Fossil Forest Member of the Fruitland Formation (Fig. 2), likely representing deposition during avulsion of a large stream or river. Disarticulation of the rostral and ventral portions of the skull, along with the chaotic distribution of other postcranial remains at the site, suggests a period of sub-aerial exposure followed by relatively little transport during deposition based on their close association. Nearly all postcranial remains at the site were eroded prior to discovery, consisting only of scattered axial and appendicular fragments.

Materials and Methods

Field methods

Collection of DMNH EPV.132300 occurred in the Bisti/De-Na-Zin-Wilderness under Bureau of Land Management permit NM14009S.

Phylogenetic methods

We utilized the phylogenetic matrix of Prieto-Márquez et al. (2018) to assess the relationships of Parasaurolophus cyrtocristatus and its congeners in light of the new information described here. Modifications to the original matrix included adding six new characters for a total of 286 characters and modified 36 from the original character codings (see Supplemental Material for complete record of changes and the new characters). Figure S1 shows the placement of the measurements used for character 286, the ratio of the posterior skull roof length to crest thickness, with the resulting measurements found in the Supplemental Material.

All analyses were performed in the program PAUP 4.0a164 using character ordering as specified in Prieto-Márquez et al. (2018)—except for the six characters added in our study, which are all unordered—heuristic search with simple stepwise addition, branch swapping via TBR, and holding 25 topologies per replicate. Iguanodon bernissartensis was the designated outgroup with the remaining 61 taxa considered ingroup. Bremer Decay Indices were calculated in PAUP using the same heuristic parameters as listed above.

Systematic paleontology

Dinosauria Owen, 1842

Ornithischia Seeley, 1888

Ornithopoda Marsh, 1881

Iguanodontia Baur, 1891 sensu Sereno, 1986 urn:lsid:zoobank.org:act:07C25E64-

6C15-40C3-8A61-A6378D8494AD

Hadrosauridae Cope, 1870 sensu Prieto-Márquez, 2010

urn:lsid:zoobank.org:act:59C23C38-0AAC-4734-81A0-F38B024AFC9F

Lambeosaurinae Parks, 1923 urn:lsid:zoobank.org:act:9ACD7CF5-9014-43FF-8C71-57CF9EBEC204

Parasaurolophini Brett-Surman, 1989 urn:lsid:zoobank.org:pub:54A1A824-214D-42C5-8626-D1479D3C8538

Parasaurolophus Parks, 1922 urn:lsid:zoobank.org:act:A287F206-E5B2-48D1-B5EF-86B25B48FA97

Revised diagnosis (§, characters derived from phylogenetic analysis; ⥉, character found in this study)

Lambeosaurine hadrosaurid taxon with the following unique traits: dorsal premaxillae extending posterodorsally over skull roof to form paired tubular chambers for narial canal (Evans, Reisz & Dupuis, 2007); common median chamber covered by lateral premaxillary process (Evans, Reisz & Dupuis, 2007); nasals extending posterodorsally nearly to end of crest in order to form hollow tubular chambers of narial canal (Evans, Reisz & Dupuis, 2007); jugal with anterior process that possesses large, dorsally directed lacrimal finger and an indentation posterior that accepts corresponding finger on the lacrimal⥉; rostral apex of rostral process of jugal reduced to a short process, only slightly thinner rostrally and ending abruptly§ (Prieto-Márquez, 2010; Prieto-Márquez et al., 2018); frontonasal platform extended posterodorsally to underlie crest (lengthens through ontogeny) (Sullivan & Williamson, 1999; Evans, Reisz & Dupuis, 2007); frontonasal platform thickened and steeply angled (Evans, Reisz & Dupuis, 2007); precerebral region of frontal short, anterior processes of conjoined frontals meet at a widely obtuse angle in dorsal view and median cleft is poorly developed (Evans, Reisz & Dupuis, 2007); olfactory depression of frontal is offset ventrally from roof of cerebral fossa (Evans, Reisz & Dupuis, 2007).

Additional unique combination of traits that differentiate Parasaurolophus from other hadrosaurids include: jugal has extremely deep posterior constriction, with a ratio greater than 1.35 (mean ratio of 1.43)§ (Prieto-Márquez, 2010; Prieto-Márquez et al., 2018); dorsal margin of infratemporal fenestra as wide as, or narrower than, quadrate cotylus and squamosal ramus of postorbital relatively short over infratemporal fenestra§ (Evans & Reisz, 2007; Prieto-Márquez et al., 2018); development of deltoid ridge dorsoventrally deep and anteroposteriorly long, with a well demarcated ventral margin§ (Prieto-Márquez, 2010; Prieto-Márquez et al., 2018); overall proportions of humerus (measured as a ratio between the total length and the width of the lateral surface of the proximal end of the humerus) relatively short and stocky, ratio less than 4.25 (mean ratio of 3.85)§ (modified from Weishampel, Norman & Grigorescu (1993), Prieto-Márquez et al. (2018)); length of ulna relative to its dorsoventral thickness (measured at mid-shaft) ratio length/width less than 10§ (Prieto-Márquez, 2010; Prieto-Márquez et al., 2018); supraacetabular crest of ilium projects lateroventrally to overlap totally or at least half of lateral ridge of the posterior prominence of ischiadic peduncle§ (modified from Horner, Weishampel & Forster (2004), Prieto-Márquez et al. (2018)).

Remarks

Sullivan & Williamson (1999) most recently revised the diagnosis of Parasaurolophus and Evans, Reisz & Dupuis (2007) provided additions to that diagnosis. In the above diagnosis, characters related to the crest were described individually in order to show how the elements contributed to overall shape of the crest. We removed reference to the length the crest previously mentioned in Sullivan & Williamson (1999) because that is recognized as an ontogenetic feature (Evans, Reisz & Dupuis, 2007; Farke et al., 2013). Additionally, unambiguous synapomorphies derived from the phylogenetic analysis performed in this study were included in the diagnosis (Prieto-Márquez et al., 2018 and references therein). These synapomorphies were included in either the apomorphic or differential diagnosis if, after being traced on the phylogeny produced in this study, they were exclusively found within the Parasaurolophus clade or were shared with other lambeosaurine taxa, respectively. Prior diagnoses (Sullivan & Williamson, 1999; Evans, Reisz & Dupuis, 2007) included the prefrontal lapping onto the anteroventral surface of the premaxilla, but we removed that trait from this diagnosis because that is a feature of all lambeosaurine hadrosaurids.

Parasaurolophus cyrtocristatus (Ostrom, 1961) urn:lsid:zoobank.org:act:842E6EA5-236F-4FE6-A406-86949A68792A

Holotype

FMNH P-27393: Partial skull composed of nearly complete right premaxilla, posterior portion of left premaxilla, mostly complete right nasal, skull roof, partial braincase, and mostly complete postcranial skeleton

Type locality, horizon, and geologic age

As reported in Ostrom (1961, p. 575), “Fruitland [F]ormation…near Coal Creek, eight miles southeast of Tsaya, McKinley County, New Mexico.” Specific locality information has been the topic of speculation by several authors (Wolberg et al., 1988; Sullivan & Williamson, 1999; Sullivan & Lucas, 2014). Consensus among these studies suggests an error in Sternberg’s original handwritten notes, and the locality is instead eight miles northeast of Tsaya, placing it within the bounds of the Fossil Forest Research Natural Area (RNA). Numerous historic quarries in the Fossil Forest RNA have been documented and may include the type locality of P. cyrtocristatus (Hunt, 1991; Sullivan & Lucas, 2014). Sullivan & Lucas (2014) further extrapolate, based on previous mapping reports (Brown, 1983; Strobell et al., 1985), that the type locality of P. cyrtocristatus is actually located in the lower Kirtland Formation rather than the Fossil Forest Member of the Fruitland Formation. However, in light of disagreement over the exact nature of the contact between the Fruitland and Kirtland as discussed above, and firsthand investigations of the Fossil Forest RNA by JJWS, we are confident that nearly all exposures within the Fossil Forest RNA are within the upper Fossil Forest Member of the Fruitland Formation. Placement within the Fossil Forest RNA (Fig. 2) is based on the work of Sullivan and Williamson (fig. 3, 1999).

Referred Materials

DMNH EPV.132300: Juvenile individual with a partial skull composed of posterior portion of premaxillae, complete nasals, right lacrimal, complete skull roof and braincase, partial left dentary, incomplete left surangular, fragmentary ribs.

Referred material locality, horizon, and geologic age

DMNH Loc. 7047, Fossil Forest Member of the Fruitland Formation in the Bisti/Den-Na-Zin Wilderness, San Juan County, northwestern New Mexico. The locality is located within the upper third of the Fossil Forest Member, dated to the Upper Campanian, stratigraphically lower than Ash 2 of Fassett & Steiner (1997) dated to 75.02 ± 0.04 Ma (Fassett & Heizler, 2017). Specific locality information is reposited at the Denver Museum of Nature & Science, available to qualified researchers upon request.

Revised diagnosis (⥉, character found in this study)

Species with the following unique traits: Long common median chamber present that is equal in dorsoventral breadth to the bounding narial tracts⥉; lateral premaxillary process has a caudoventral extension that projects to nearly the edge of the crest⥉; crest curved sharply posterior to skull (Ostrom, 1961); postcotyloid process of squamosal expands proximoanteriorly and tapers to form a notch of ~49° within the quadrate cotylus⥉; median ramus of squamosals inset at their ventral articulation, forming a pyramidal grotto underlain by the supraoccipitals⥉.

Additionally, this species possesses a unique suite of traits that differentiates from other taxa: Crest with smooth sides (Sullivan & Williamson, 1999); crest height to skull roof length ratio ~1⥉; absence of the external mandibular muscle scar on anterior squamosal⥉; median ramus of squamosals risen to be subvertical⥉.

Description

DMNH EPV.132300 consists of a partial skull excellently preserved.

Premaxilla—The premaxillae of Parasaurolophus cyrtocristatus are almost fully known between the type specimen FMNH P-27393 (Fig. 3) and the newly referred specimen DMNH EPV.132300 (Figs. 4 and 5). Along the oral margin of the right premaxilla, FMNH P-27393 preserves the remains of conical denticles as in other hadrosaurids. A short distance from the oral margin the narial fenestra (bony naris) originates as a rounded opening that terminates in a pinched form posteriorly. The anterior margin of the narial fenestra marks the split between the dorsal and lateral processes of the premaxilla whereas the posterior terminus of the narial fenestra marks the coalescence of the two processes to form the enclosed bony narial passage.

Figure 3 Skull of Parasaurolophus cyrtocristatus (FMNH P-27393).

(A) Left lateral view and (B) right lateral view. Scale bar equals 10 cm.

Figure 4 Skull of Parasaurolophus cyrtocristatus (DMNH EPV.132300).

(A) Photograph of right lateral side; (B) illustration of right lateral side; (C) photograph of left lateral side; and (D) illustration of left lateral side. Abbreviations: Bso, Basioccipital; Bsp, Basisphenoid; Exo, Exoccipital; F, Frontal; La, Lacrimal; Lsp, Laterosphenoid; Na, Nasal; Osp, Orbitosphenoid; Pa, Parietal; Pmd, premaxilla dorsal process; Pml, premaxilla lateral process; Po, Postorbital; Pr, Prootic; Prf, Prefrontal; Ps, Presphenoid; Sq, Squamosal.

Figure 5 Skull of Parasaurolophus cyrtocristatus (DMNH EPV.132300).

(A) Photograph of ventral skull; (B) illustration of ventral skull; (C) photograph of posterior skull; (D) illustration of posterior skull. Abbreviations: Bso, Basioccipital; Bsp, Basisphenoid; Exo, Exoccipital; F, Frontal; Fm, Foramen Magnum; La, Lacrimal; Lsp, Laterosphenoid; Na, Nasal; Pa, Parietal; Pmd, premaxilla dorsal process; Pml, premaxilla lateral process; Pmx, Premaxilla; Po, Postorbital; Pr, Prootic; Prf, Prefrontal; Ps, Presphenoid; So, Supraoccipital; Sq, Squamosal.

In P. cyrtocristatus, FMNH P-27393, the dorsal processes propagate up the dorsal surface of the skull in a nearly straight to subtly concave line, which differs from the straight to slightly more convex growth of the premaxillae seen on P. walkeri (ROM 768) (Fig. 1). Given the less than ideal state of preservation of P-27393 we currently attribute the difference in morphology to taphonomic deformation. There appears to be no evidence of plastic deformation on the type specimen because oval-shaped sandstone steinkerns of the narial passage are observable where the now eroded left anterior premaxilla resided. These sandstone tubes are uniformly straight and have a consistent measurement of 5 cm in maximum width and 3.5 cm perpendicular to that maximum. Therefore, we regard this morphology as original and distinctive.

Uncrushed open crest sutures on the referred juvenile specimen DMNH EPV.132300 clearly demonstrate that the dorsal and lateral premaxillary processes form the bulk of the cranial crest. The dorsal processes form individual tubes that create median and ventral septa to keep the airways isolated. Following the dorsal processes posteriorly reveals that they form the entire dorsal, posterior, and a minor ventral portion of the supracranial crest (Fig. 6). Observation of a crest cross-section on FMNH P-27393 (Fig. 6) shows that during the dorsal progression of the dorsal processes of the premaxillae, they remain in a simple tubular structure with relatively thin walls, whereas on the posterior to ventral transition these processes develop a thick outer wall. A change in external texture and crest form corresponds to the regions of pachyossfication. More specifically, the external texture on the thinner-walled tube is smooth and the thickened tube appears roughened. Additionally, the posteroventral tip of the crest on DMNH EPV.132300 possesses a slightly “pulled” shape that is distinct from the smooth curvature of the dorsal crest. At their terminus, these premaxillary dorsal processes reside between the nasals as in all other hadrosaurids.

Figure 6 Detailed photographs of the Parasaurolophus cyrtocristatus crest.

(A) FMNH P-27393 in right lateral view; (B) cross section of the posterior cranial crest on FMNH P-27393; (C) posterior crest of FMNH P-27393 in right lateral view; (D) supracranial portion of crest on DMNH EPV.132300 with lines demarcating skull elements; (E) supracranial portion of crest on DMNH EPV.132300 without lines demarcating skull elements. Note that letters z and z′ denote the homologous points on the FMNH P-27393 crest. Abbreviations: Na, Nasal; Pmd, Premaxilla dorsal process; Pml, Premaxilla lateral process; pnf, premaxilla-nasal fontanelle. Scale bars equal 5 cm.

Complementing the dorsal processes, the lateral premaxillary processes rise along the crest as well, but instead of forming tubes, these elements form a strap that passes on the medial side of the prefrontals above the skull roof. They continue as laminar processes, covering the oblong opening created by the tubular dorsal premaxillary processes and the nasals, referred to as the common median chamber. The bone texture of the lateral processes on FMNH P-27393 has a flat conchoidal shape that is not seen on DMNH EPV.132300. The latter specimen perfectly preserves the lateral processes, showing that on their terminus they possess a long tab that extends ventrally to onlap the posterior portion of the dorsal processes (Fig. 6). Additionally, as seen on DMNH EPV.132300, the lateral processes do not fully converge with the dorsal processes at the terminus of the crest, leaving a fontanelle as reported for the juvenile-aged Parasaurolophus sp. by Farke et al. (2013). FMNH P-27393 does not show clearly the fontanelle, although an eroded portion in the same region as the DMNH EPV.132300 fontanelle is seen on the right side; therefore, it seems that this feature closes on Parasaurolophus cyrtocristatus through ontogeny, but a better preserved, more mature individual is needed to confirm this.

Nasals—The anterior half of the nasals can be observed on DMNH EPV.132300 underlying the premaxillary processes. They share a long straight articulation with the lacrimal along their lateral edge (Fig. 7). While supported ventrally by the frontonasal platform, the nasals also contact the prefrontals laterally (Fig. 4), sharing an extended articulation with this element and the lacrimal anterior to the platform. Though not well preserved, it appears on FMNH P-27393 that the paired nasals form the ventral portion of the distinctive tubular narial crest of parasaurolophs, and indeed DMNH EPV.132300 confirms that the nasal forms approximately 80% of lower tubes at an earlier ontogenetic stage. Upon emerging from the fronto-nasal platform, these paired elements form complete tubes and propagate posteriorly along the base of the crest until they articulate with the dorsal premaxillary process a short distance from the posterior crest terminus (Figs. 4, 5, and 6). As in all other hadrosaurids, the nasals form a V-shaped opening into which the dorsal premaxillary processes insert. A cross-section of the crest on FMNH P-27393 shows that the nasal tubes were divided from each counterpart by a median septum. Just anterior to the frontonasal platform the nasal tubes converge into a single rounded chute.

Figure 7 Anterolateral view of Parasaurolophus cyrtocristatus DMNH EPV.132300 showing the relationship of the nasal, lacrimal, and prefrontal.

Abbreviations: La, lacrimal; Na, Nasal; Pf, Prefrontal; Pm, Premaxilla.

Lacrimal—A mostly complete lacrimal is present on DMNH EPV.132300, but lacking in the P. cyrtocristatus type specimen. It has the same general morphology of other hadrosaurid lacrimals in being triangular in lateral view, but also possesses the synapomorphic large notch on the ventral surface that accepts the corresponding process from the jugal as seen on P. walkeri (ROM 768) and P. tubicen (NMMNH P-25100). Medially, the lacrimal shares a long articular surface with the nasal (Fig. 7). It articulates with the prefrontal posterodorsally through a “V”-shaped notch in the lacrimal that accepts a long corresponding process of the latter element. Ventral to the prefrontal articulation on the posterior side of the lacrimal is the opening for the lacrimal foramen. Unlike the widening posterior lacrimal face seen in some saurolophines (e.g., Gryposaurus (Gates & Sampson, 2007), Acristavus (Gates et al., 2011)), that of DMNH EPV.132300 maintains a single width throughout its posterior height. A difference in lacrimal morphology can be seen in P. walkeri such that this element of ROM 768 protrudes posteriorly into the orbit, whereas that of DMNH EPV.132300 is smoothly arching. Given that only a single specimen is known of P. walkeri, individual variation could explain the difference in lacrimal morphology seen in this species versus P. cyrtocristatus, but until more specimens are acquired, we consider this to be a species-specific trait. That of P. tubicen (NMMNH P-25100) seems to correspond in shape to DMNH EPV.132300.

Prefrontal—Among lambeosaurines, the prefrontal forms the base of the crest complex, forming a supporting structure on the ventrolateral section of the crest (Lull & Wright, 1942). Parasaurolophus cyrtocristatus (DMNH EPV.132300 and FMNH P-27393), and other Parasaurolophus species more generally, follow this pattern, but also exaggerate the size of the prefrontal compared to lambeosaurins (Evans & Reisz, 2007; Evans, 2010). In DMNH EPV.132300, the prefrontal is tightly sutured to the postorbital laterally. There is a small foramen with a posteriorly directed furrow just dorsal to the postorbital suture near the orbital rim. As one follows the orbital margin from the postorbital anteriorly on the prefrontal, the rugose texture becomes less pronounced in conjunction with a decreasing prominence of the brow.

Articulation with the lacrimal occurs along the “V”-shaped process on the anterior margin. Medial to this articulation, the prefrontal flattens to contact the ventrolateral side of the nasals as they project along the base of the crest complex. This arrangement continues posterodorsally as the crest rises beyond the skull roof. The prefrontal in all species of Parasaurolophus extends posterodorsally farther than in other lambeosaurines.

Frontal—Only the posterior half of the frontals are visible on DMNH EPV.132300 and FMNH P-27393 because of articulation with the crest-forming elements and sediment matrix in-filling. Though it is not seen on any P. cyrtocristatus specimen, the frontonasal platform likely conforms to that of both P. walkeri and P. tubicen in consisting of a highly angled striated platform (Wiman, 1931; Evans, Reisz & Dupuis, 2007; Evans, Bavington & Campione, 2009). The anterior damage to the frontal of FMNH P-27393 mentioned by Sullivan & Williamson (1999) is erosion of the dorsal premaxillary process ventral tube margin. Evans, Reisz & Dupuis (2007) noted that the body of Parasaurolophus frontals is extremely anteroposteriorly shortened, declaring this trait a synapomorphy of the genus. The frontal body of the type specimen and DMNH EPV.132300 differ only in that the latter specimen is highly domed, a trait seen in other juvenile lambeosaurs (Evans, Forster & Reisz, 2005). Posteriorly, the frontonasal platform is observed as paired, flat struts that ventrally support the nasals of the crest spanning either a short distance along the base of the crest as in DMNH EPV.132300 (relative length of 0.07) or nearly half the length of the ventral crest in FMNH P-27393 (relative length of 0.19, see Supplemental Material for measurements), likely as a consequence of growth stage. One of the most obvious similarities between members of Parasaurolophus and the clade’s sister taxon Charonosaurus is the striated, highly angled, and posterodorsally expanding frontal platform (Godefroit, Zan & Jin, 2001). Although it does appear, based on Godefroit, Zan & Jin (2001) figure 4A, that the frontal platform is mediolaterally broader than that in Parasaurolophus. More detailed comparisons are needed of this element between the two taxa.

Postorbital—The tripartite postorbital has an anteromedial arm that robustly contacts the prefrontal, frontal, and minorly on the parietal. A second ventrally directed arm articulates with the jugal, and the third, posteriorly oriented arm, overlaps the anterior process of the squamosal. Along the postorbital contribution to the orbit the DMNH EPV.132300 postorbital is bubble-shaped, being much more rounded dorsally than the anterior portion of the orbit composed of the prefrontals. This region of the orbital rim also possesses minor rugose texturing. A foramen 5 mm wide resides near the orbital rim. Overlap of the postorbital onto the squamosal is accomplished through a wide scarf joint that is weakly bipartite as in the Corythosaurus specimen AMNH 5386 and Hypacrosaurus altispinus (Evans, 2010). Other hadrosaurs have two equally long processes including lambeosaurs (e.g., Velafrons; (Gates et al., 2007) or saurolophines generally (Prieto-Márquez, 2010; Gates et al., 2011)), but the articular facets can also be jagged as in H. stebingeri (MOR 553S 7-27-2-93).

Parietal—Following the form of other lambeosaurines, this element is slightly hourglass shaped articulating with the frontal anteriorly along a long suture. There are unobserved sutures to other elements, presumably the prootic and exoccipital, that have been overgrown with bone. Posteriorly, the parietal of Parasaurolophus cyrtocristatus takes on a more distinct form by increasing its height to be higher than any other lambeosaur except P. tubicen. The sagittal crest begins at the midpoint of the parietal, following posteriorly to bisect the squamosals (Fig. 8).

Figure 8 Parasaurolophus cyrtocristatus DMNH EPV.132300 squamosal.

(A) Right laterodorsal, (B) and posterior view with a close-up of the quadrate cotylus notch in (C) FMNH P-27393 and (D) DMNH EPV.132300. Abbreviations: bptp, basipterygoid processes; Bso, Basioccipital; bst, basitubera; elsq, elevated posterior squamosal; Exo, Exoccipital; Fr, Frontal; itf, infratemporal fenestra; mr, median ramus of squamosal; Pa, Parietal; Po, Postorbital; pop, paroccipital process; prcp, precotyloid process; postcp, postcotyloid process; qc, quadrate cotylus; Spo, Supraoccipital; Sq, squamosal. Scale bar equals 5 cm.

Squamosals—Being one of the more distinctive elements in the DMNH EPV.132300 cranium, the squamosals are best described in congruence with multiple elements. They articulate with the postorbitals anteriorly via scarf joint; medially, the parietal separates the paired squamosals with a narrow wedge that becomes extremely thin posteriorly and ventrally, a feature shared among lambeosaurs (Prieto-Márquez, 2010); and ventrally the squamosals are underlain by the supraoccipital. Of minor consequence to the gross morphology is the scarf joint between the postcotyloid process and paroccipital process of the exoccipital-opisthotic.

As in many other lambeosaurines, the dorsal aspect of the squamosal median processes are broadened into a partial canopy over the dorsal temporal fenestrae. However, unlike species such as Hypacrosaurus stebingeri (MOR 553S 7-27-2-93), H. alitspinus (Evans, 2010), Corythosaurus casuarius (AMNH 5386), and Olorotitan arharensis (Godefroit, Bolotsky & Bolotsky, 2012) in which the broadened median processes obscure a larger portion of the dorsal temporal fenestra, within Parasaurolophus cyrtocristatus (both DMNH EPV.132300 and FMNH P-27393) the squamosals adhere to parietals that are elevated well above the level of the anterior parietals. As such, each median process creates more of a posterior wall than shading the dorsal temporal fenestra (Fig. 8). Godefroit, Alifanov & Bolotsky (2004) describe the median region of the Aralosaurus squamosal rising well-above the paroccipital process, but the condition observed, at least in their figure 5, is still well below the height observed in P. cyrtocristatus. Charonosaurus (Godefroit, Zan & Jin, 2001) does not preserve this region of the squamosals. Interestingly, P. tubicen shares the trait of the greatly elevated squamosals and parietals with P. cyrtocristatus (PMU.R1250 and NMMNH P-25100), but the trait is not present in P. walkeri (ROM 768). Additionally, when viewed posteriorly, it becomes apparent that the dorsal tip to the median process along with the sandwiched parietal extend posteriorly further than the proximal main squamosal body, forming an anteroventrally-oriented angle that terminates at the supraoccipital. Since the latter element is flat, the resulting morphology is a distinct pyramidal grotto that is unique to P. cyrtocristatus (Fig. 8).

Two other interesting traits are found on the lateral surface of the squamosal (Fig. 8). First, the postcotyloid process is expanded dramatically at its proximal region, so much so, that the quadrate cotylus is nearly walled along its medial border. This feature is seen both on DMNH EPV.132300 and FMNH P-27393 to nearly the same angulation (42 and 49 degrees, respectively, at their widest point). Expansion of the postcotyloid process tapers ventrally in a straight line. P. tubicen also shares an expanded, but tapering, postcotyloid process, although the condition in this species is less dramatic that in P. cyrtocristatus. The same region is obscured in P. walkeri (ROM 768). Outside of lambeosaurines, an analogous expansion can be found in several saurolophs, notably Prosaurolophus, Kundurosaurus (Godefroit, Bolotsky & Lauters, 2012), and Naashoibitosaurus. In each of these taxa, the postcotyloid process expansion forms a squared-off tab instead of a proximal expansion as described above. Edmontosaurus annectens shares a subdued version of the postcotyloid tab. This trait seems to be a synapomorphy of those hadrosaurids most closely related to Prosaurolophus. A second interesting trait of the Parasaurolophus cyrtocristatus squamosal is that the deep triangular scar for external mandibular musculature seen in other hadrosaurids (Gates & Sampson, 2007) is absent in all species of Parasaurolophus.

Braincase—The braincase of DMNH EPV.132300, is well preserved, although the sutures between individual elements are difficult to discern (Figs. 4, 5, and 9), making an accurate deconstruction of this element complex impossible without the assistance of internal inter-element observations aided by commuted tomography. However, Fig. 9 shows the various structures present on the braincase of DMNH EPV.132300 that are identifiable irrespective of element sutures.

Figure 9 Parasaurolophus cyrtocristatus DMNH EPV.132300 braincase.

(A) Right lateral view. (B) Ventral view. Abbreviations: ap, alar process; Bso, Basioccipital; bptp, basipterygoid processes; Bsp, Basisphenoid; bspg, basisphenoid groove; bst, basitubera; cno, cranial nerve opening; Exo, Exoccipital; ibsl, interbasisphenoid lamina; Osp, Orbitosphenoid; pop, paroccipital process; qc, quadrate cotylus; vd, Vidian Canal. Scale bars equal 5 cm.

Basisphenoid-Parasphenoid complex—From the incomplete parasphenoid (Figs. 4, 5, and 9), the rounded anterior process becomes slightly concave when followed posteriorly. Simultaneously with the changes in the presphenoid process, two sharply defined ridges form laterally, which widen during their posterior progression. As they descend the basipterygoid processes the sharp ridge dissolves again into rounded pendant-shaped bones (Figs. 4, 5, and 9). Between the two basipterygoid is a lamina that delineates the divide between a triangular concave region anteriorly and a shear wall posteriorly. Absent from this lamina is a small ventral projection that can be seen on other taxa including Hypacrosaurus altispinus (Evans, 2010), Hypacrosaurus stebingeri (MOR 553S 7-27-2-93), Corythosaurus (Evans, 2010), and Amurosaurus (Godefroit, Bolotsky & Itterbeeck, 2004). At the base of the wall is a deep groove that runs posteriorly from this point (approximately mid-length of the basisphenoid) to the posterior terminus of the bone, with the groove extending slightly onto the basioccipital. Other taxa that possess the same deep fossa are H. stebingeri (MOR 553S 7-27-2-93) and Charonosaurus (Godefroit, Zan & Jin, 2001).

Alar processes are minutely developed on this taxon, being only tiny teardrop-shaped projections just anterior to the Vidian Canal. This trait is shared with Parasaurolophus tubicen (NMMNH P-25100), whereas other lambeosaurines have more developed alar processes (e.g., Charonosaurus (Godefroit, Zan & Jin, 2001), Hypacrosaurus altispinus (Evans, 2010), Olorotitan (Godefroit, Bolotsky & Bolotsky, 2012), and Amurosaurus (Godefroit, Bolotsky & Itterbeeck, 2004)).

Fusion of the basisphenoid with the basioccipital is clearly discernible, being demarcated by the basisphenoid cupping the anterior tubercles of the basioccipital. A combination of the central groove and broad articulation of these two elements gives the basisphenoid cups a “mouse-ear” appearance.

Basioccipital—The anterior morphology of the basioccipital (Figs. 4, 5, 8, and 9) is predicated by the deep fossa that depresses the posterior basisphenoid and projects this depression onto the basioccipital. As such, the laterally positioned basitubera are elevated relative to the median depression as well as to the basioccipital body just posterior to the basitubera. The posterior margin of the basioccipital that contributes to the occipital condyle is only slightly enlarged compared to the central portion of the element. In posterior view, this articular facet has an ovoid heart shape, and comprises the central-most floor of the foramen magnum. Overall, the gross morphology of the basioccipital does not differ from that seen in other lambeosaurine hadrosaurs.

Exoccipital-opisthotic complex—Situated posterodorsally to the basioccipital, the exoccipital (Figs. 4, 5, 8, and 9) composes the remainder of the pendant-shaped foramen magnum. Two blunt processes rest on the posterodorsal surface of the basioccipital and project posteriorly approximately 1 cm beyond the latter element. The fused opisthotic can be seen descending lateroventrally from the laterodorsal surface of the exoccipital. A separation between the opisthotic and the paroccipital process of the squamosal shows that the opisthotic descended slightly beyond the ventral tip of the squamosal. This elongated condition differs dramatically from the rounded spatula shape of the opisthotics in Charonosaurus (Godefroit, Zan & Jin, 2001). Hypacrosaurus stebingeri (MOR 553S 7-27-2-93) possesses opisthotics that rise well above the level of the squamosals. This condition is not possible in Parasaurolophus cyrtocristatus because of the vaulted squamosals.

Just dorsal to the foramen magnum, the DMNH EPV.132300 exoccipital forms an anteroventrally inclined shelf that meets the supraoccipital along its dorsal margin, similar to the depressed exoccipital of Olorotitan (Godefroit, Bolotsky & Bolotsky, 2012). Hypacrosaurus stebingeri (MOR 553S 7-27-2-93), H. altispinus (Evans, 2010), Velafrons (Gates et al., 2007), and Amurosaurus (Godefroit, Bolotsky & Itterbeeck, 2004) do not possess the same morphology, but instead have a shorter margin between the foramen magnum and the supraoccipital, which decreases the length and angle of descent on this portion of the exoccipital.

Supraoccipital—Exposure of the supraoccipital (Figs. 5 and 8) is limited to a small triangular shelf dorsal to the exoccipitals and ventral to the squamosals. In DMNH EPV.132300, the combination of the squamosals and the supraoccipital form a tall pyramidal grotto that differs from the condition seen in Parasaurolophus tubicen (PMU.R1250, Sullivan & Williamson, 1999, fig. 8) and other lambeosaurines.

Orbitosphenoid—This small cuniform bone articulates with the frontal medially and dorsally, and with the laterosphenoid posteriorly and ventrally (Figs. 4 and 9). It houses the opening for cranial nerve II, but is otherwise a simple, laminar element.

Laterosphenoid—Occupying a central position on the braincase, the left laterosphenoid of DMNH EPV.132300 is preserved in its entirety (Figs. 4 and 9). Dorsally, the laterosphenoid meets the parietal along a linear, anteroposteriorly directed suture. The capitate process of the laterosphenoid articulates dorsolaterally with the postorbital medial to the midpoint of the central lamina of the postorbital. Anteriorly, the laterosphenoid meets the lamina of the orbitosphenoid in a simple, linear suture, continuing anterodorsally to form a simple suture with the frontal. Posteriorly, the laterosphenoid meets the prootic in a difficult to discern suture near the exit of CN VII. Posteroventrally, the laterosphenoid contributes to the anterior and dorsal walls of the trigeminal foramen (CN V) a large, ovoid opening set within a deep recess. A deeply set groove extends anteriorly across the lateral surface of the laterosphenoid for passage of the opthalmic branch of CN V, partially roofed by a sharp crest on the laterosphenoid. A shallower groove extends anteroventrally from the trigeminal recess for passage of the maxillary and mandibular branches of CN V, partially bordered by the ventral process of the laterosphenoid.

Prootic—The prootic (Figs. 4, 5, and 9) contributes to the posterodorsal portion of the lateral braincase wall, extending ventrally from its dorsal contact with the parietal to form a laterally convex lamina. The prootic forms the posterior margin of the trigeminal foramen and the anterior and dorsal margins of the fenestra vestibuli. Between each of these openings, the prootic is pierced by the fenestra for CN VII immediately posterior to an anteroventrally-directed crest near the suture with the laterosphenoid. The prootic meets the exoccipital posteriorly.

Dentary—Only the proximal half of the right dentary (Figs. 10A and 10B) is preserved with DMNH EPV.132300. Overall, this partial element is typical of hadrosaurid iguanodontians. Narrow alveolar sulci extend from the approximate midportion of the element posterior to end within the rostrocaudal length of the basal coronoid process. Throughout the preserved length of the dentary, the articular facet for the angular is prominent medially, yet hidden in lateral view. The coronoid process rises nearly vertically from the dentary body with a dorsal margin that expands both rostrally and caudally, unlike the same region on Brachylophosaurus that only expands rostrally (Prieto-Márquez, 2005; Gates et al., 2011). The taxon Charonosaurus also has limited caudal expansion of the dorsal coronoid process, and additionally differs from P. cyrtocristatus in having a rostroventrally oriented dorsal-most surface of this process (Godefroit, Zan & Jin, 2001). A straight ventral margin is featured on the preserved portion of the dentary.

Figure 10 Mandibular elements of Parasaurolophus cyrtocristatus DMNH EPV.132300.

(A) Right dentary in lateral view; (B) right dentary in medial view; (C) right surangular in dorsal view; (D) right surangular in ventral view. Scale bar equals 5 cm.

Surangular—This element was found eroded on the surface, as such, the more delicate processes are not preserved. As shown in Figs. 10C and 10D, the main surangular body is typical of iguanodontians, yet the more derived features of hadrosaurids are missing.

Results

Phylogenetic analysis

A strict consensus tree constructed from the 72 most parsimonious topologies obtained in our phylogenetic analysis posited Parasaurolophus cyrtocristatus as sister taxon to Parasaurolophus tubicen (Fig. 11), supported by two unambiguous synapomorphies (281. Squamosals risen to be subvertical; and 283. Triangular external mandibular muscle scar anterior to precotyloid process absent). The unambiguous synapomorphies uniting all species of Parasaurolophus exclusive of other hadrosaurids are characters 103 (Rostral apex of the rostral process of the jugal reduced to a short process, only slightly thinner rostrally and ending abruptly), 111 (Relative depth of the jugal caudal and rostral constrictions (in adults) rostral constriction region located between the rostral and postorbital processes; caudal constriction region located between the postorbital process and the caudoventral flange extremely deep with a ratio greater than 1.35), 128 (Narrowing of the dorsal region of the infratemporal fenestra being as wide as or narrower than quadrate cotylus), 216 (Development of the scapula deltoid ridge dorsoventrally deep and craniocaudally long, with a well demarcated ventral margin), 220 (Overall proportions of the humerus (measured as a ratio between the total length and the width of the lateral surface of the proximal end of the humerus relatively short and stocky humerus, ratio less than 4.25), 221 (Length of the ulna relative to its dorsoventral thickness (measured at mid-shaft) ratio length/width less than 10), and 234 (Development of the lateroventral projection of the supraacetabular crest of the ilium projects lateroventrally to overlap totally or at least half of the lateral ridge of the caudal prominence of the ischiadic peduncle). Other clades of lambeosaurine hadrosaurids remain in the same construction as in Prieto-Márquez et al. (2018). Bremer decay indices show that the characters and codings in this analysis support a robust framework for more basal iguanodontians, whereas the clades of hadrosaurids suffer from less robust support. The Brachylophosaurini clade (Gates et al., 2011) and the clade containing the genera Augustynolophus and Saurolophus (members of the Saurolophini clade (Prieto-Márquez et al., 2014)) both have Bremer decay values greater than 4. Subclades throughout the Parasaurolophini (Brett-Surman, 1989) are supported as other lambeosaurine clades with Bremer values of 2 throughout.

Figure 11 Strict consensus phylogenetic tree derived from 72 most parsimonious trees derived from analysis in this study.

Tree scores: tree length 1044, CI 0.399, RI 0.777, RC 0.310. Underlined numbers at each node are the bootstrap values, whereas standalone numbers designate the Bremer Decay Index for that branch. Note that the Bremer indices of 4 and greater are approximations because the analysis was stopped after retaining 4.2 million trees. Grey box encloses the Parasaurolophini clade. Characters were ordered as in Prieto-Márquez et al. (2018).

Discussion

Phylogenetic and biogeographic considerations

Parasaurolophus cyrtocristatus was erected as a short-crested species of Parasaurolophus by Ostrom (1961), but subsequently suggested to be a sexual dimorph of P. tubicen by Hopson (1975). Most recent systematic revisions of the genus Parasaurolophus have discussed the validity of Parasaurolophus cyrtocristatus and at least tentatively considered the taxon valid (Sullivan & Williamson, 1999; Horner, Weishampel & Forster, 2004; Evans, 2007, 2010; Prieto-Márquez, 2010). Confusingly, Williamson (2000) argued that it is most parsimonious to regard P. cyrtocristatus as a subjective junior synonym of P. tubicen, yet at the same time referred BYU 2467 and UCMP 143270 to P. cyrtocristatus later in the same paragraph. In this study, new material affirms the validity of P. cyrtocristatus. The results of the phylogenetic analysis conducted in this study are the first to suggest that P. tubicen and P. cyrtocristatus are more closely related to each other than either is to P. walkeri. Previously, the two ‘long-crested’ species were hypothesized as a clade (Evans & Reisz, 2007; Evans, 2010; Godefroit, Bolotsky & Bolotsky, 2012; Prieto-Márquez et al., 2018) or as members of a monophyletic clade with “Charonosaurus” jiayinensis (Xing et al., 2014a, 2014b) in which P. cyrtocristatus is the sister taxon to these other parasaurolophs.

Interestingly, the results of this phylogenetic analysis are more consistent with the paleobiogeographic and stratigraphic occurrences of Parasaurolophus cyrtocristatus and P. tubicen, which are both found in northern New Mexico—2000 kilometers south of the Alberta occurrences of P. walkeri—and in considerably younger geologic formations separated by a maximum 2.5 million years (Gates, Prieto-Márquez & Zanno, 2012). Parasaurolophus walkeri is known only from the basal portion of the Dinosaur Park Formation (Ryan & Evans, 2005; Evans, Bavington & Campione, 2009), which means that there is a temporal separation of approximately 2 million years between this taxon and P. cyrtocristatus, whereas there is a maximum 4.5 million years between the former taxon and P. tubicen.

Another proposed parasauroloph from North America is Adelolophus hutchisoni, a species identified from an isolated maxilla that was discovered approximately 200 m from the base of the Wahweap Formation, making it the oldest lambeosaurine currently known from North America (Gates et al., 2013, 2014; Holroyd & Hutchison, 2016). Given its fragmentary nature, phylogenetic or biogeographic comparisons between this species and the other parasaurolophs are extremely limited, and we therefore refrain from making further inferences about the relationships of Adelolophus.

Comparison to the Kaiparowits Formation Taxon

The first specimens of Parasaurolophus described from the Kaiparowits Formation possess a drastically arched crest as seen in P. cyrtocristatus. Despite the authors of the original description of BYU 2467 expressly opting out of providing a species referral, largely because BYU 2467 is only a poorly preserved partial crest (Weishampel & Jenson, 1979). A second specimen, UCMP 143270 (Figs. 12 and 13), constitutes a well preserved and complete crest with articulated upper cranium. Sullivan & Williamson (1999) and Williamson (2000) referred both BYU 2467 and UCMP 143270 to P. cyrtocristatus, based on crest curvature. New preparation and study of UCMP 143270 together with the discovery of DMNH EPV.132300 gives the opportunity to test the identification of the Kaiparowits taxon with the newly described characteristics of P. cyrtocristatus since more recent studies have shown that crest curvature is allometric.

Figure 12 Photographs and illustrations of Parasaurolophus cf. cyrtocristatus UCMP 143270.

(A) Photograph of left lateral side; (B) illustration of left lateral side; (C) photograph of right lateral side; and (D) illustration of right lateral side. Abbreviations: Bso, Basioccipital; Bsp, Basisphenoid; Exo, Exoccipital; F, Frontal; La, Lacrimal; Lsp, Laterosphenoid; Na, Nasal; Osp, Orbitosphenoid; Pa, Parietal; Pmd, premaxilla dorsal process; Pml, premaxilla lateral process; Po, Postorbital; Pr, Prootic; Prf, Prefrontal; Ps, Presphenoid; Sq, Squamosal.

Figure 13 Photographs and illustrations of Parasaurolophus cf. cyrtocristatus UCMP 143270.

(A) Photograph of ventral skull; (B) illustration of ventral skull; (C) photograph of posterior skull; (D) illustration of posterior skull. Abbreviations: Bso, Basioccipital; Bsp, Basisphenoid; Exo, Exoccipital; FM, Foramen Magnum; La, Lacrimal; Lsp, Laterosphenoid; Na, Nasal; Osp, Orbitosphenoid; Pmd, premaxilla dorsal process; Pml, premaxilla lateral process; Pmx, Premaxilla; Po, Postorbital; Pr, Prootic; Prf, Prefrontal; Ps, Presphenoid; So, Supraoccipital; Sq, Squamosal.

UCMP 143270, illustrated here for the first time, possesses the raised squamosals of P. tubicen and P. cyrtocristatus (Fig. 12). In caudal view one can see the pyramidal grotto between the squamosals and the supraoccipital (Fig. 13). The postcotyloid process seems to show a slight expansion into the quadrate cotylus, but both the right and left sides are broken so it is unclear if the expansion is as narrow as P. cyrtocristatus or if it is wider as in P. tubicen. The sutures of the crest are clearly visible allowing its construction to be understood in similar detail to DMNH EPV.132300. The size of the lateral premaxillary process is most comparable to that of P. cyrtocristatus (Fig. 12), which is certainly different than the narrower crest of P. walkeri or the broader crest of P. tubicen (Fig. 1). Finally, the lateral premaxillary process does not show a posteroventrally directed flange as in DMNH EPV.132300 (Fig. 12). This could be the result of simple breakage or individual variation, or could be an indication that this specimen is not referrable to P. cyrtocristatus.

After compiling evidence, and considering the possible confounding effects of individual and ontogenetic variation, we concur with Williamson (2000) and tentatively identify the Kaiparowits taxon (as exemplified by BYU 2467 and UCMP 143270) as P. cyrtocristatus based on the general osteological make-up of the crest, and the presence of the pyramidal grotto. If this identification holds after further detailed investigation of the increasing sample of mature individuals from the same beds, UCMP 143270 will provide an important poorly known ontogimorph of this genus and outstanding complement to DMNH EPV.132300 for comparisons of braincase anatomy. Additionally, this taxonomic identification would add to a growing body of evidence that the Kaiparowits Formation has a close biotic relationship with both the Fruitland and Kirtland formations of northern New Mexico (e.g., ankylosaurids (Sullivan, 1999; Arbour et al., 2014; Wiersma & Irmis, 2018) and ceratopsids (Sampson & Loewen, 2010; Fowler & Freedman Fowler, 2020).

Crest composition and taphonomy

Another important implication of this study is that DMNH EPV.132300 and UCMP 143270 are the first specimens to unequivocally show the detailed anatomy of the external crest surfaces, which has been uncertain and problematic for phylogenetic studies of this taxon. Previous studies of crest anatomy have been based primarily on the poorly preserved and/or badly crushed and difficult to interpret type specimens of the three Parasaurolophus species. Most of these studies posit an extremely small lateral premaxillary process and a small laterally exposed nasal of uncertain boundaries that does not contribute significantly to the ventral margin of the crest (Weishampel, 1981) or that forms a very small contribution to the base of the crest (Russell, 1946; Sullivan & Williamson, 1999). Evans, Bavington & Campione (2009) reviewed the various hypotheses and interpretations of crest composition in Parasaurolophus walkeri, noting that the composition of the crest in any specimen is uncertain due to ambiguities of preservation.

These two new, remarkably preserved specimens permit a detailed understanding of external crest anatomy in a species of Parasaurolophus for the first time and confirm that the paired nasals form the basal tubes along four-fifths the ventral length of the supracranial crest, and that the dorsal premaxillary process forms the narial tubes throughout the entire dorsal, posterior, and slight ventral portions of the crest. Based on this new interpretation, the composition of the crest is in fact more consistent with patterns of crest construction seen in other lambeosaurines (e.g., Corythosaurus and Lambeosaurus) in which the dorsal premaxillary process forms the crest dorsally, the lateral premaxillary process forms a large lateral portion of the crest ventral to the dorsal process, and the nasal has a significant contribution to the posterior region and base. Interestingly, this new knowledge of crest anatomy in Parasaurolophus cyrtocristatus, which we hypothesize is the general construction present in P. walkeri and P. tubicen, closely resembles a hypothesis of crest construction presented by Wilfarth (1939), where he took an evolutionary approach to interpreting origin of crest structure in Parasaurolophus tubicen.

An interesting trend in the preservation of most Parasaurolophus cranial material is the prevalence of articulated skull roofs. Specimens that consist solely or largely of this portion of cranial material include the type and referred specimens of P. tubicen, the type and referred specimens of P. cyrtocristatus, as well as the published specimens from the Kaiparowits Formation BYU 2467 and UCMP 143270. This represents over 85% of crest-yielding material of Parasaurolophus are articulated skull roofs. Even in specimens described by Evans, Reisz & Dupuis (2007) and Evans, Bavington & Campione (2009), the highly eroded braincases are articulated, including a juvenile specimen. Growth of the large cranial crest may require that the skull become reinforced from an early ontogenetic stage, which in part may explain the prevalence of these skull sections being preserved with such frequency. The greatly enlarged crests would certainly have put relatively more stress on the joints of the skull roof and braincase in Parasaurolophus relative to other hadrosaurid taxa. Increased interdigitation in the skull roof is observed in other amniote taxa as a response to increased stress (Herring, 2008, 1974), and this could explain the propensity for the tops of skulls to maintain cohesion and be preferentially preserved together in Parasaurolophus.

Conclusions

Ornamental cranial crests are some of the most distinguishing features of hadrosaurid dinosaurs. Consequently, the morphological distinctions between crests of various species have been thoroughly studied, often times to the detriment of identifying additional characteristics across the range of skull elements that can be used to diagnose species. Species of the genus Parasaurolophus have suffered this bias, and none as markedly as P. cyrtocristatus. There is little doubt that the source of this tendency stems from the poor preservation of the type and only specimen as well as from the distinct, sharply curving crest which departs from the longer, straighter crest of its congeners P. walkeri and P. tubicen.

DMNH EPV.132300 is a well-preserved new specimen of P. cyrtocristatus that was discovered in the Fossil Forest Member of the Fruitland Formation, the same set of rocks from which the type specimen was presumably excavated, which allows for a reassessment of the species-specific traits that make the taxon novel. Aside from the sharply curving crest (which will remain as part of the diagnosis until other, more mature specimens are recovered), we have identified a narrow notch in the quadrate cotylus of the squamosal that is formed by an expansion of the anterodorsal and posterodorsal cotylus corners, a pyramidal-shaped grotto between the squamosals and the supraoccipitals, and the lateral process of the premaxilla with a small posteroventral projecting finger along the side of the crest.

An additional six features that we identify, such as vertically projecting squamosals and an absent mandibular muscle scar on the anterolateral squamosal, are used in a new phylogenetic analysis that posits a close relationship between P. tubicen and P. cyrtocristatus. This is the first phylogenetic analysis to our knowledge that suggests this sister-taxon relationship. In our analysis, P. walkeri forms the basal member of the Parasaurolophus clade, which is consistent with its lower stratigraphic position.

Proper diagnosis of each Parasaurolophus species is required to fully understand the paleobiological implications of sexual selection, crest allometry, and natural selection on the overall changes observed in skull architecture. This clade of hadrosaurs is one of the iconic dinosaurs, yet so much of their evolution and paleobiology is unknown, only to be revealed by more detailed anatomical studies and the discovery of new specimens.

Supplemental Information

Supplemental Information 1 Phylogenetic matrix of characters and codings used in this analysis.

Click here for additional data file.

Supplemental Information 2 List of phylogenetic characters used in this study and changes to original matrix.

Click here for additional data file.

Supplemental Information 3 Diagram showing locations of measurements to calculate the ratio of crest thickness used in character 286 of the phylogenetic matrix.

Click here for additional data file.

Supplemental Information 4 Raw measurements of Parasaurolophus crests and skulls for phylogenetic matrix.

Click here for additional data file.

Supplemental Information 5 Tree file with all of the most parsimonious trees obtained from the analyses.

Click here for additional data file.

Supplemental Information 6 Nexus file of the consensus tree obtained from the phylogenetic analysis.

Click here for additional data file.

DMNH EPV.132300 was discovered by Erin Spear and collected with the guidance and support of Phil Gensler (BLM Regional Paleontologist) and the Farmington BLM Field Office. Special thanks to the amazing DMNS volunteer field crew, especially C. Hank Woolley, Larkin McCormack, Doug Shore, Andrew Spear, and Erin Spear for their assistance in collecting the specimen. Preparation was expertly accomplished by Becky Garfield under the supervision of Mike Getty, Natie Toth, and Salvador Bastien. We thank Peter Makovicky and Bill Simpson for access to Field Museum collections. We thank P. Holroyd and Mark Goodwin for the loan of the UCMP specimen to DCE for preparation and study, and Ian Morrison (ROM) for preparation of the specimen. Andrew McDonald, Albert Prieto-Márquez, and Pascal Godefroit provided valuable comments that greatly improved the manuscript. Caleb Brown provided information on specimens under his care. Finally, we would like to respectfully acknowledge that the fossils from the San Juan Basin were collected on the traditional lands of the Diné and Puebloan people, and fossils from the Kaiparowits Basin were collected on the traditional lands of the Kaibab Paiute, Hopi, Ute, Dine, and Puebloan peoples.

Institutional abbreviations

BYU Brigham Young University, Provo, UT, USA

DMNH Denver Museum of Nature & Science, Denver, CO, USA

FMNH Field Museum of Natural History, Chicago, IL, USA

NMMNH New Mexico Museum of Natural History, Albuquerque, NM, USA

ROM Royal Ontario Museum, Toronto, ON, CA

UMNH Natural History Museum of Utah, Salt Lake City, UT, USA

Additional Information and Declarations

Competing Interests

Author Contributions

Field Study Permissions

Data Availability

The authors declare that they have no competing interests.

Terry A. Gates conceived and designed the experiments, performed the experiments, analyzed the data, prepared figures and/or tables, authored or reviewed drafts of the paper, and approved the final draft.

David C. Evans conceived and designed the experiments, performed the experiments, analyzed the data, prepared figures and/or tables, authored or reviewed drafts of the paper, and approved the final draft.

Joseph J.W. Sertich conceived and designed the experiments, performed the experiments, analyzed the data, prepared figures and/or tables, authored or reviewed drafts of the paper, and approved the final draft.

The following information was supplied relating to field study approvals (i.e., approving body and any reference numbers):

Field collection was conducted in the Bisti/De-Na-Sin-Wilderness under permit NM14009S from the Bureau of Land Management.

The following information was supplied regarding data availability:

Parasaurolophus cyrtocristatus specimens described here are accessioned at the Field Museum of Natural History under specimen number FMNH P-27393, the Denver Museum of Natural Science under specimen number DMNS EVP.132300, and the University of California Berkeley Museum of Paleontology under specimen number UCMP 143270.

The Mesquite phylogenetic matrix and character list, an illustration of the measurements required for character 286, raw measurement, and tree files are available in the Supplemental Files.

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
