# Peer review of "Description and rediagnosis of the crested hadrosaurid (Ornithopoda) dinosaur Parasaurolophus cyrtocristatus on the basis of new cranial remains"

_PeerJ, doi:10.7717/peerj.10669_

## Round 0.1 · original submission · Minor Revisions

The reviewers are positive. Nevertheless, they raise a few issues, which should be addressed, and offer suggestions for improvement, which should be taken into account. Please, check the author of Iguanodontia (as far as I know, it's Baur, not Dollo).

Together with your unmarked revised manuscript, provide a marked-up copy as well as a document explaining how you have addressed each of the points raised by the reviewers.

·

Basic reporting

No comment.

Experimental design

No comment.

Validity of the findings

No comment.

Additional comments

Reviewer: Andrew T. McDonald

This is an excellent description of an important new specimen that clarifies long-standing ambiguity regarding the crest of Parasaurolophus and sets the stage for future work on the extensive Kaiparowits sample. Very little revision is necessary. I would suggest adding two brief thoughts to the Discussion.

1. In the "Phylogenetic and biogeographic considerations" section, perhaps mention that your results, with P. cyrtocristatus and P. tubicen as sister taxa, indicate that P. tubicen and P. walkeri evolved their elongated crests convergently, but that P. tubicen is more similar to P. cyrtocristatus in having a dorsoventrally broader crest. As you note in the next section, the crest of P. cyrtocristatus is broader than that of P. walkeri but narrower than that of P. tubicen.

2. In the "Comparison to the Kaiparowits Formation Taxon" section, in the last paragraph in which you tentatively identify the Kaiparowits specimens as P. cyrtocristatus, you could mention that other recent studies have found additional close relationships between the ornithischians of the Kaiparowits Formation and Fruitland/Kirtland formations. These include the ankylosaurines Akainacephalus (Wiersma & Irmis 2018) and Nodocephalosaurus (Sullivan 1999), and the chasmosaurines Utahceratops (Sampson et al. 2010) and the lineage of Pentaceratops-Navajoceratops-Terminocavus (Fowler & Freedman Fowler 2020).

·

Basic reporting

This is a great contribution to our understanding of lambeosaurine hadrosaurid anatomy and evolution, derived from a beautiful and exquisitely preserved specimen. The text is very clear and concise, and the descriptions are particularly insightful and contain a wealth of detailed information on the comparative osteology of Parasaurolophus (except in the case of the dentary, but I comment on that below). The figures are excellent, of great quality and very informative, enhancing the data provided by this study (I, however, have some minor remarks on the orientation of some of the elements, but see below). I agree with pretty much all the results and conclusions presented here, which are supported by sufficient evidence as documented here by the authors. Thus, I certainly recommend publication of this study, with the minor revisions detailed below.

Experimental design

Field Methods

Line 164: “Collection of DMNH EPV.132300 occurred in the Bisti-De-Na-Sin-Wilderness in
collected under Bureau of Land Management permit NM14009S.” I would delete “in collected”.

Phylogenetic Methods

How many replicates in the heuristic search?

Validity of the findings

Revised diagnosis of P. cyrtocristatus:
Line 293-294: “Preorbital portion of premaxillae straight, not convex”. I am looking at the lateral view of the holotype skull of P. walkeri, ROM 768, and I am also seeing a straight preorbital portion of the premaxillae. Thus, I would remove this character from the diagnosis of P. cyrtocristatus. What appears different to me in P. cyrtocristatus is that the preorbital region of the premaxillae (as shown in FMNH P-27393) is slightly, very gently, concave.

Line 294: “long common median chamber present that is equal in dorso-ventral breadth
to the bounding narial tracts”. Since other combinations such as “caudoventral” and “proximoanteriorly” have been written without a hyphen, I would write “dorsoventral” rather than “dorso-ventral”.

Description:

Lines 320-322: “In P. cyrtocristatus, FMNH P-27393, the dorsal processes propagate up the
dorsal surface of the skull in an autapomorphic straight line, which differs from the more
convex growth of the premaxillae seen on P. walkeri (ROM 768) (Fig. 1).” I see that happening when not restricting this condition to the preorbital region.

Lines 462-464: “However, unlike species such as Hypacrosaurus stebingeri (MOR 553S 7-27-2-93), Hypacrosaurus alitspinus (Evans, 2010), Corythosaurus (AMNH 5386), and Olorotitan
(Godefroit, Bolotsky & Bolotsky, 2012)”. But the species of Corythosaurus and Olorotitan are not given here, just the genera. Also, it should be written H. altispinus.

Line 570: “Laterosphenoid--“. There should be an “m dash” rather than two hyphens.

Line 593: The description of the dentary is too simple and lack details, particularly in comparison with the detailed description of the numerous skull elements. I would like to see the dentary described in more detail. Also, rather than “narrow tooth rows” (line 596) I would say “narrow alveoli” or “narrow alveolar sulci”. Overall it would seem as if the dentary was described in haste (the period is even missing at the end of the paragraph).

Results of the phylogenetic analysis

Lines 611-613: “The unambiguous synapomorphies uniting all species of Parasaurolophus exclusive of other hadrosaurids are characters 103, 111, 128, 216, 220, 221, and 234.” To prevent the reader from having to leave this page and search what characters are 103, 111, 128, 216, 220, 221, and 234, I would prefer to see these characters spelled out here. After all, space is not a problem in PeerJ.

Discussion

It would be helpful to have an additional figure showing the stratigraphic position of all Parasaurolophus spp. specimens, that are discussed in the text.

Figures

Fig. 1. In the caption, instead of just “skulls”, I’d write “holotype skulls” to be more precise.

Fig. 6. A and B should better be placed vertically, as if looking at the skull mounted. I don’t think that by being shown sideways, the images can be displayed in substantially greater size.

Fig. 10. These elements are shown in an odd orientation. Why not display them in anatomical position?

For the sake of completeness, why not add a figure with BYU 2467? Since it is discussed alone with UCMP 143270, which it is beautifully illustrated here.

Additional comments

No further comments

·

Basic reporting

This is a nice paper describing interesting new material of Parasaurolophus, with interesting discussions about Parasaurolophus taxonomy and palaeogeography. I have just sfew questions/comments listed below:

Line 211: "frontonasal platform extended posterodorsally to underlie crest" This character is also present in Charonosaurus. At all, comparisons with Charonosaurus (regarded as the sister-taxon for Parasaurolophus) are extremely limited in this paper. More comparisons with this closely-related taxon would be helpful.

Line 321. Why do you estimate that the straight process of the premaxilla in P. cyrtocristatus is the autapomorphic condition?

Line 351 (7) should be (Fig. 7)?

Line 360: “The anterior half of the nasals can be observed on DMNH EPV.132300 to share a long straight articulation with the lacrimal posterolaterally”. Are you sure about it??? If so, please illustrate it accordingly.

Line 363: “sharing an extended articulation with this element and the lacrimal anterior to the platform”. Idem, not in adequation with the illustrations!

Line 370: “(Figs. PCSKULL and 7)”. What is PCSKULL?

Line 381: “Medially, the lacrimal shares a long articular surface with the nasal.”. Idem, requires illustration.

Line 400: replace “lambeosaurins” by “lambeosaurines”.

Line 420: shortening of the frontal also observed in Charonosaurus.

Experimental design

Taphonomy: Your explanation about the preservation of articulated skull roofs in Parasaurolophus is interesting, but I have another question: is the postcranial material of P. cyrtocristatus really rare, or is it just an excavation/publication/description bias (nobody likes to excavate or describe disarticulated postcranial material)?

Validity of the findings

As already mentionned above, additional illustration of the lacrimal-nasal articulation should be provided.

In general, more references towards the figures should be helpful to clarify the text.

---

## Round 0.2 · Minor Revisions

Correct the author of Iguanodontia when you get a chance.

In addition, the Section Editor, Mark Young, noticed the following:

> There are a few minor typos in the text, the kind which aren't always picked up on in proof stage.
> Also, the cladogram-space search was not exhaustive. If it would be possible for the authors to do a TNT advanced methods search, then TBR the results (bbreak) that would be be more thorough.
> Also, the authors use "trees" at times where they mean "cladogram" or "topology".

Please address all of these in a minor revision.

·

Basic reporting

I think that the authors adequately responded to the questions/concerns of the three reviewers, so the ms can be published as such in Peer J.

Experimental design

Idem

Validity of the findings

Idem

Additional comments

Idem

---

## Round 0.3 · accepted · Accept

As indicated in my first decision, the author of Iguanodontia is Baur (1891). Also, please remove the parentheses in Systematic Paleontology. I accept your MS now but urge you to make these edits before publication.